# Hereditary Hyperferritinemia Cataract Syndrome: Ferritin L Gene and Physiopathology behind the Disease—Report of New Cases

**DOI:** 10.3390/ijms22115451

**Published:** 2021-05-21

**Authors:** Ferran Celma Nos, Gonzalo Hernández, Xènia Ferrer-Cortès, Ines Hernandez-Rodriguez, Begoña Navarro-Almenzar, José Luis Fuster, Mar Bermúdez Cortés, Santiago Pérez-Montero, Cristian Tornador, Mayka Sanchez

**Affiliations:** 1Iron Metabolism: Regulation and Diseases, Department of Basic Sciences, Universitat Internacional de Catalunya (UIC), 08195 Sant Cugat del Vallès, Spain; fcelma@uic.es (F.C.N.); ghernandezv@uic.es (G.H.); xferrerc@uic.es (X.F.-C.); 2BloodGenetics S.L. Diagnostics in Inherited Blood Diseases, 08950 Esplugues de Llobregat, Spain; sperez@bloodgenetics.com (S.P.-M.); Ctornador@bloodgenetics.com (C.T.); 3Hematology Service, University Hospital Germans Trias i Pujol (HGTiP), Institut Català d’Oncología (ICO), 08916 Badalona, Spain; agnesrh@iconcologia.net; 4Hematology and Hemotherapy Service, Clinic University Hospital Virgen de la Arrixaca, Instituto Murciano de Investigación Biosanitaria (IMIB), 30120 Murcia, Spain; b.almenzar@gmail.com; 5Pediatric OncoHematology Service, Clinic University Hospital Virgen de la Arrixaca, Instituto Murciano de Investigación Biosanitaria (IMIB), 30120 Murcia, Spain; josel.fuster@carm.es (J.L.F.); mariam.bermudez2@carm.es (M.B.C.)

**Keywords:** hereditary hyperferritinemia cataract syndrome, HHCS, serum ferritin, *FTL* gene, cataracts, hyperferritinemia, IRE, IRP

## Abstract

Hereditary hyperferritinemia-cataract syndrome (HHCS) is a rare disease characterized by high serum ferritin levels, congenital bilateral cataracts, and the absence of tissue iron overload. This disorder is produced by mutations in the iron responsive element (IRE) located in the 5′ untranslated regions (UTR) of the light ferritin (*FTL*) gene. A canonical IRE is a mRNA structure that interacts with the iron regulatory proteins (IRP1 and IRP2) to post-transcriptionally regulate the expression of proteins related to iron metabolism. Ferritin L and H are the proteins responsible for iron storage and intracellular distribution. Mutations in the FTL IRE abrogate the interaction of FTL mRNA with the IRPs, and de-repress the expression of FTL protein. Subsequently, there is an overproduction of ferritin that accumulates in serum (hyperferritinemia) and excess ferritin precipitates in the lens, producing cataracts. To illustrate this disease, we report two new families affected with hereditary hyperferritinemia-cataract syndrome with previous known mutations. In the diagnosis of congenital bilateral cataracts, HHCS should be taken into consideration and, therefore, it is important to test serum ferritin levels in patients with cataracts.

## 1. Introduction

Hereditary hyperferritinemia-cataract syndrome (HHCS) (OMIM #600886) is a rare disease characterized by high serum ferritin levels, congenital bilateral cataracts, and the absence of tissue iron overload. It was first described in 1995 independently by two researcher teams in France and Italy as an autosomal dominant inherited disease [1,2]. These original manuscripts reported one and two families, respectively, with the previously mentioned clinical features. As no known disease could describe their patient’s phenotype, they propose this new genetic disorder.

Beaumont described a three-generation family with visual acuity impairment since young age with deposits in all layers of both lens and an elevated level of circulating ferritin with no other hematological or biochemical abnormalities [2].

On the other hand, Girelli described two north Italian families: Family 1 presented congenital nuclear cataracts and high serum ferritin levels. The proband and his daughter were treated as hereditary hemochromatosis with phlebotomies until they developed microcytic anemia. In the second family, the proband presented bilateral cataracts; however, clinical doctors did not detect elevated serum ferritin levels until his third decade of life; this could be attributed to the fact that the patient was a blood donor [1].

The prevalence of the disease is 1 in 200,000 individuals. However, this number is probably underestimated because the disorder is often misdiagnosed as hereditary hemochromatosis due to hyperferritinemia, despite the absence of iron overload [3]. Actually, investigators often find out new cases with previously reported mutations, as in the case of Ferro and collaborators that described an Italian family [4]; for this reason, it is an important differential diagnosis.

## 2. Genetics of HHCS

HHCS is due to mutations in the iron responsive element (IRE) located in the 5′ UTR of the Ferritin L mRNA (FTL). At least 47 mutations have been described in *FTL* gene as causative of HHCS, including 36 single mutations, 9 deletions, and 2 insertion-deletions (Figure 1 and Appendix A). Some authors attempted to find out a relationship between the disease severity and the position of the mutation in the IRE structures [5,6,7]. Thus, Luscieti and collaborators correlated higher serum ferritin levels in patients with mutations located in the hexanucleotide loop or bulge C than in patients with mutations located in less critical regions as the upper or lower stems of the IRE. Most of the mutations causing HHCS are usually present in heterozygous state due to the dominant inheritance of the disease. However, there are few reported cases with mutations in homozygous state [7,8].

Traditionally, the HHCS mutations were named following a nomenclature containing the name of the city where they have been described and the number of the mutational position counting from the transcription initiation site as +1; this is referred to as the traditional nomenclature (i.e., +40A>G Paris-1). The official HGVS nomenclature numbered the mutations considering +1 the A in the ATG of the starting codon (methionine 1, ATG), and thus mutations on the 5′ UTR have negative numbering (i.e., c.-160A>G is the same as the +40A>G Paris-1 mutation).

## 3. Molecular Mechanism behind HHCS

Ferritin is a highly conserved protein in evolution, and it is responsible for cellular iron storage. Cellular ferritin is composed of two different subunits, H-Ferritin and L-Ferritin, which are encoded by two different genes: *FTH* and *FTL* genes, respectively. A canonical ferritin is constituted by the assembly of 24 H- and L-ferritin subunits, forming a chest-like structure that can store thousands of iron atoms. H-Ferritin is the heavy subunit and presents ferroxidase activity converting ferrous iron (Fe^+2^) into ferric iron (Fe^+3^), allowing its storage. Whereas, L-Ferritin is part of the ferritin core and it is in charge of ferrous iron release when needed. Furthermore, serum ferritin is mainly constituted by L-ferritin and is partially glycosylated [9].

The expression of both ferritins is regulated post-transcriptionally by the iron regulatory proteins (IRP1 and IRP2) through its binding to the iron-responsive elements (IREs), a conserved hairpin-like motif, located in the 5′ untranslated region (UTR) of ferritin mRNAs [10]. A canonical IRE structure is composed of a six-nucleotide apical loop 5′-CAGUGN-3′ (N denotes A, C or U) on a stem of five paired nucleotides, a small asymmetrical bulge with an unpaired cytosine on the 5′ strand of the stem, and an additional lower stem of variable length [11].

In iron replete conditions, the IRP1 assembles Fe-S clusters and does not interact with IREs, while IRP2 is degraded. However, in iron-deficient conditions, IRPs bind with IREs to modify gene expression (Figure 2). Depending on the IRE location in the UTR, the binding with IRP has different effects. Thus, if the IRE is located in 3′ UTR, the IRP binding causes mRNA stabilization, but if the IRE is in the 5′ UTR, as in the case of *FTL* gene, the IRP binding produces protein translational repression. Mutations in the 5′ IRE of FTL mRNA lead to a loss of interaction between IRPs and the FTL IRE, and a loss of translation repression by them; consequently, an excessive level of ferritin is produced and can be detected in the serum (Figure 2).

## 4. Physiopathology of HHCS

Two main hypotheses have been postulated for the cause of the cataract formation in HHCS. The first hypothesis is related to iron homeostasis, and it is postulated that the overproduction of L-ferritin may increase free iron and reactive oxygen species (ROS) with concurrent oxidative damage to the lens that produces loss of transparency [13]. However, the facts that L-Ferritin does not directly bind iron and that the lens-diffracting crystals are iron-poor contradict this possible explanation [14].

The second and most accepted hypothesis refers to the loss of proper lens transparency due to the accumulation of large amounts of crystal deposits formed by aggregated and iron-poor L-ferritin, as demonstrated by Brooks and collaborators [14]

Uncontrolled production of serum ferritin due to the mutations in the L-ferritin IRE causes an excess of cellular ferritin that is secreted to the serum, where it is detected as hyperferritinemia. The excess of ferritin precipitates in the lens cortex, forming numerous small punctate, white breadcrumb-like opacities. The deposits are light-diffracting and impede the correct focus by the patient. In this disease, cataracts are bilateral and usually appear at early ages, however, the visual impairment is slight and patients may not need a surgical intervention until they reach adult age [14,15]. Moreover, the morphology of the HHCS cataracts is highly distinctive from other cataract types with scattered central and peripheral cystic flecks in the cortex and nucleus and small crystalline aggregates. Thus, some authors postulate the possibility of diagnosing HHCS, analyzing the cataract morphology as the main feature [3].

## 5. Treatment for HHCS

The HHCS patients have no clinical symptoms other than bilateral cataract. L-ferritin deposits in the lens are growing constantly but slowly, and patients diagnosed with HHCS are treated by undergoing surgical intervention when they suffer visual impairment due to the cataracts formation, usually in the adult stage. A suspicion of HHCS and a positive genetic testing will alert for ophthalmologists follow-up of the patient and would most probably result in cataract early detection and therefore a possible intervention at an earlier age.

HHCS is usually misdiagnosed as hereditary hemochromatosis due to the detection of hyperferritinemia that is a common biochemical sign between both diseases. The importance of a differential diagnosis, including genetic diagnostic, is highlighted to avoid potential damage treatments as venesections or phlebotomies that are indeed counterproductive as it will end up causing iron-deficiency anemia in HHCS patients [8]. In the case of HHCS, phlebotomies are definitely not the proper treatment to apply as these patients do not have iron overload. Their hyperferritinemia does not reflect an excess of body iron but rather an impairment of L-ferritin post-transcriptional regulation.

## 6. Report of Two New Families with HHCS

### 6.1. Family A

Proband III.2 from family A (Figure 3 family A) is a 38-year-old woman referred to the Hematology Service at the University Hospital Germans Trias i Pujol (HGTiP) in Barcelona because of unexplained hyperferritinemia. Serum Ferritin levels were 1143 ng/mL, while serum iron and transferrin saturation levels were normal (Table 1). Additional biochemical and hematological data are listed in Table 1. Hereditary Hemochromatosis (HH) genetic studies were negative with no pathogenic mutation found in the HH-related genes. Liver magnetic resonance showed normal levels of hepatic iron. Other relatives, the mother and the maternal uncle, reported to have cataracts and hyperferritinemia; in addition, a first cousin of the mother has hyperferritinemia. Interestingly, three family members died at a young age as a result of ischemic events (cerebral vascular accident or acute myocardial infarction) as reflected in the pedigree, so the proband is currently under study for thrombophilia.

Genetic studies performed on proband (A-III.2) showed the presence of a heterozygous point mutation c.-160A>G located in the 5′ FTL IRE. This variant consists of a substitution in the conserved apical hexanucleotide loop (CAGUGU) (Figure 1). This mutation is indeed the first pathogenic alteration reported for HHCS and described by Beaumont in 1995; the traditional nomenclature referred to it as +40A>G Paris1 mutation.

### 6.2. Family B

Proband III.1 from family B (Figure 3) is a 44-year-old woman that has presented with hyperferritinemia since 8 years, so she was referred to the Hematology Service in the University Hospital Virgen de la Arrixaca in Murcia. The patient had no other significant clinical history but endometriosis. Her serum ferritin level was 919 ng/mL; serum iron and transferrin saturation levels were normal (Table 1). Additional biochemical and hematological data are listed in Table 1. She was tested for hereditary hemochromatosis HFE gene mutation, and she was found to be a carrier of the H63D variant in HFE gene (this genetic finding could not explain her hyperferritinemia). She had normal deposits of liver iron (<20 μmol/g) in magnetic resonance imaging. During anamnesis, the patient reported having visual problems, so she was directed to an ophthalmology service, where she was diagnosed with cataracts and underwent cataract surgery at the age of 44 years. At that moment, she was interrogated about her family history. Proband’s mother (B-II.2) also was HFE H63D heterozygous and had hyperferritinemia, so she was misdiagnosed as HH and treated with phlebotomies. Proband’s mother (B-II.2) had also undergone cataract surgery when she was younger. Proband’s maternal aunt (B-II.3) has hyperferritinemia and cataracts but she has not undergone cataract surgery. The proband also has a brother (B-III.2) with hyperferritinemia and congenital cataracts that has been operated, and two nieces, one had normal ferritin levels and no cataracts (IV.2) and an older niece (IV.1) who was also genetically studied because of hyperferritinemia and operated congenital cataracts (see below).

In family B, patient IV.1 is the niece of the proband (B-III.1) (Figure 3 family B), she was 10 year old when she attended the hematology consult due to hyperferritinemia. Her serum ferritin levels were 931 ng/mL and transferrin saturation was normal. She was diagnosed with cataracts at the age of 10 years old. Additional biochemical and hematological data are listed in Table 1.

After genetic testing by Sanger sequencing of the *FTL* gene on the proband B-III.1 and by a NGS gene panel on the patient B-IV.1 from the same family, both patients were found to be heterozygous for the NM_00146.3:c.[-167C>T];[=] mutation in *FTL* gene, also known as the +33C>T Madrid/Philadelphia pathological variant that was reported for the first time by Balas and collaborators [16]. This pathogenic mutation is located in the C-bulge of the FTL IRE (Figure 1), and this is a critical residue for IRP-IRE interaction.

These two families (A and B, Figure 3) illustrate the clinical findings and the mode of inheritance (autosomal dominant) of hereditary hyperferritinemia-cataract syndrome (HHCS) and presented with mutations in the apical loop or the C-bulge of the FTL IRE, respectively.

## 7. Discussion

Hereditary hyperferritinemia-cataract syndrome (HHCS) was the first genetic disorder known to result from regulatory mutations affecting translation (Figure 2). As such, it represents a novel mechanism for cataract formation. Since its discovery in 1995, different authors have reported mutations that affect the FTL IRE and cause HHCS (Appendix A and Figure 1). Here, we describe two new families with several affected members, and identify previously reported mutations in *FTL* gene, the +40A>G Paris1 (c.-160A>G) [2] and the +33C>T Madrid/Philadelphia (c.-167C>T) [16] mutations, for family A and B, respectively. The +40A>G Paris1 mutation affected the IRE apical loop, while the +33C>T Madrid/Philadelphia mutation was located in the C-buge of the IRE. Both of these IRE positions are critical residues for IRP-IRE interaction [17]. In both families, the ferritin levels were substantially high (above 900 ng/mL), which is in agreement with the conclusions reported by Luscieti and collaborators, who demonstrated that despite all mutations in the FTL IRE increasing serum ferritin, mutations in the hexanucleotide loop or in the C-bulge lead to higher serum ferritin levels than mutations located in other IRE positions [7].

In patients with HHCS, the unique clinical abnormality is the presence of congenital cataracts. Thus, to improve the diagnosis of this disease, it is very relevant to increase awareness in ophthalmologists about the importance of testing serum ferritin levels routinely while investigating cataracts, especially in pediatric cases. On the other side, all unexplained hyperferritinemia should be referred for ophthalmological assessment, as cataracts may be asymptomatic but lead to a correct diagnosis of HHCS. The definitive confirmation of the disease will be achieved after a simple genetic test as all mutations leading to HHCS are concentrated in a concrete region comprising the 5′ UTR region of *FTL* gene.

## 8. Materials and Methods

### 8.1. Patients

All subjects gave their informed consent for inclusion in this study. The study was conducted in accordance with the Declaration of Helsinki, and the protocol was approved by the Ethics Committee of the Hospital General de Catalunya.

### 8.2. DNA Extraction, PCR Amplification and DNA Sequencing

Genomic DNA was extracted from peripheral blood using the FlexiGene DNA kit (Qiagen, Hilden, Germany) according to manufacturer’s instructions.

Ferritin L (FTL) gene region was sequenced using the Sanger method or NGS by using a gene panel. For Sanger sequencing, the FTL gene was amplified using 50 ng of genomic DNA. Descriptions of the primers and conditions have been previously published elsewhere [18]. The resulting amplification products were verified on a 2% ethidium bromide agarose gel. The purified PCR products were sequenced using a conventional Sanger method. Sequencing results were analyzed using Mutation Surveyor software (SoftGenetics LLC, State College, PA, USA).

For NGS methods, patient B-IV.1 was analysed using a targeted NGS gene panel (v15) for hereditary hemochromatosis and hyper/hypoferritinaemia (BloodGenetics panel #10010) that included the following nine genes: HFE, HFE2, HAMP, TFR2, SLC40A1, BMP6, FTL, FTH1, and GNPAT, following a protocol reported previously for a different panel [19]. Data analysis was performed using Varsome clinical software [20]. Mutations detected by NGS were confirmed by conventional Sanger sequencing.

Genetic variants are reported following the official human genome variation sequence (HGVS) nomenclature and refer to NM_000146.3 for the Homo sapiens FTL transcript variant and NP_000137.2 for the Homo sapiens FTL protein.

## Figures and Tables

**Figure 1 ijms-22-05451-f001:**
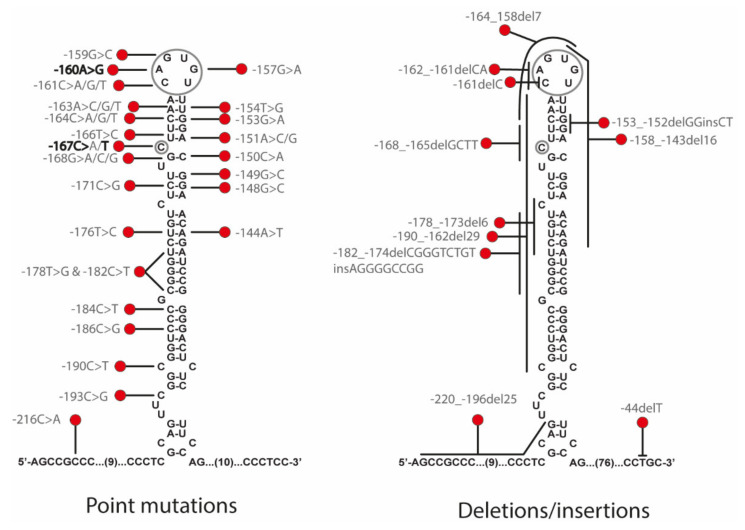
Summary of all reported mutations that cause HHCS. Point mutations are on the right and deletions/insertions are depicted on the left. A grey circle marks the C-bulge and the hexanucleotide loop. Mutations in the families from this study are marked in bold text (c.-160A>G family A and c.-167C>T family B). Genetic variants are reported following the official human genome variation sequence (HGVS) nomenclature. For more detailed information, see Appendix A.

**Figure 2 ijms-22-05451-f002:**
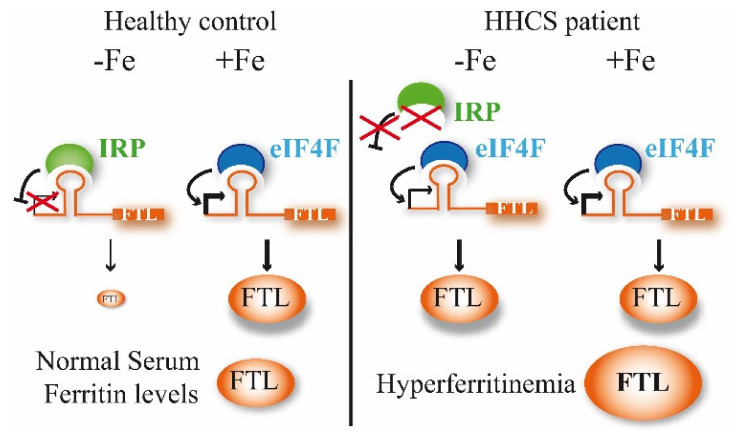
Regulation of *FTL* expression by the IRP/IRE post-transcriptional regulatory system and iron effect on *FTL* translation. IRPs can bind the IRE under low iron conditions (−Fe) forming a repressor complex for protein synthesis. Iron (+Fe) increases the binding of the translation factor eIF4F, forming an activator complex for protein synthesis. IRP and eIF4F compete for the IRE biding [12]. In patients with HHCS, mutations in the FTL IRE abrogate the IRP-IRE interaction, and overall, there is a loss of the post-transcriptional regulation, producing an excess of ferritin that can be detected in serum as hyperferritinemia.

**Figure 3 ijms-22-05451-f003:**
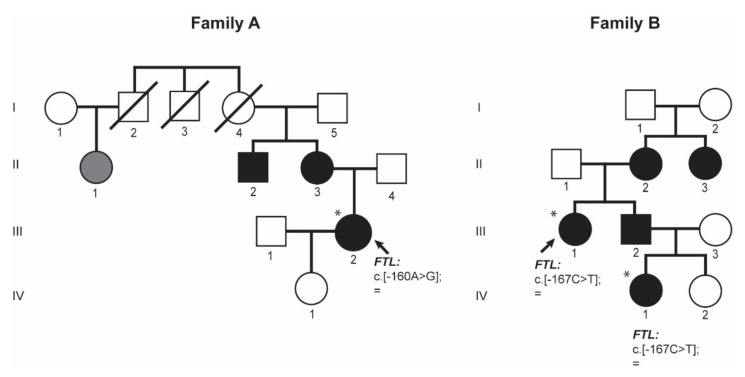
Pedigree trees for the two studied families affected by HHCS. Squares indicate males, and circles, females; slashed symbols indicate deceased individuals. Probands are pointed with an arrow. Filled symbols indicate affected individuals by genetically confirmed HHCS or by the presence of hyperferritinemia and cataracts; grey symbol denotes an individual where only hyperferritinemia has been reported. Asterisks indicate subjects with genetic studies done at BloodGenetics SL. Mutation is named according to the HGVS nomenclature.

**Table 1 ijms-22-05451-t001:** Biochemical and hematological data of the two studied families affected with HHCS.

Case	Family A	Family B	Family B	Reference Values
Patient	II.1	III.1	IV.1	-
Gender	F	F	F	-
Age at diagnosis (years)	38	44	11	-
Hb (g/dL)	13.3	12.7	14	13.5–17.5 (M) 12.1–15.1 (F)
MCV (fL)	83.3	87.9	82.6	80–95
Ferritin (ng/mL)	1143	919	931	12–300 (M) 12–200 (F)
TF sat (%)	23.0	21.9	14.7	25–50
Iron (μg/dL)	66.45	116	n.a.	49–226
*FTL* Mutation	c.-160 A>G	c.-167 C>T	c.-167 C>T	-

The following abbreviations were used: Hb, hemoglobin; MCV, mean corpuscular volume; TF sat, transferrin saturation; F, female; M, male; n.a., not available data. The mutation nomenclature used follows the HGVS guidelines.

## Data Availability

The data presented in this study are available on request from the corresponding author.

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
