# Peer review of "Hereditary Hyperferritinemia Cataract Syndrome: Ferritin L Gene and Physiopathology behind the Disease—Report of New Cases"

_ijms, 2021, doi:10.3390/ijms22115451_

Round 1
Reviewer 1 Report
It is an interesting article speak about a very rare genetics disease, that is often underdiagnosed and misdiagnosed. We ask you to insert a further paragraph on the differential diagnosis and its complexities.
considering that they describe two other Italian families, we ask to deepen the review on the cases and families already described in Italy. It is suggested to implement the bibliography on Italian cases, for example Ferro E et al. FTL c.-168G>C Mutation in Hereditary Hyperferritinemia Cataract Syndrome: A New Italian Family. Pediatr Dev Pathol. 2018 Sep-Oct;21(5):456-460. doi: 10.1177/1093526618755200. Epub 2018 Feb 9. PMID: 29426274, and to describe the most frequent variants described in Italian patients.
Author Response
IJMS/2021/05/17
REVIEWERS ANSWERS FIRST VERSION
Original Title and authors
Hereditary Hyperferritinemia Cataract Syndrome: Ferritin L Gene and Physiopathology Behind the Disease. Report of New Cases
Ferran Celma Nos 1,†, Gonzalo Hernández 1,†, Xènia Ferrer-Cortès 1,2, Ines Hernandez-Rodriguez 3, Begoña Navarro-Almenzar 4,5, José Luis Fuster 5, Mar Bermúdez Cortés 5, Santiago Pérez-Montero 2, Cristian Tornador 2 and Mayka Sanchez 1,2,*
† These authors contributed equally to this work.
Detailed responses to the reviewers’ comments and suggestions
We are grateful to the two reviewers for their thorough assessment of our paper, for their constructive input and for their overall very positive appreciation of our work. We have carefully revised our manuscript and addressed each point that was raised to our best, as detailed below. The new version of our manuscript contains the modifications as highlighted text for a rapid inspection.
Reviewer 1 Comments for the Author
It is an interesting article speak about a very rare genetics disease, that is often underdiagnosed and misdiagnosed. We ask you to insert a further paragraph on the differential diagnosis and its complexities.
Considering that they describe two other Italian families, we ask to deepen the review on the cases and families already described in Italy. It is suggested to implement the bibliography on Italian cases, for example Ferro E et al. FTL c.-168G>C Mutation in Hereditary Hyperferritinemia Cataract Syndrome: A New Italian Family. Pediatr Dev Pathol. 2018 Sep-Oct;21(5):456-460. doi: 10.1177/1093526618755200. Epub 2018 Feb 9. PMID: 29426274, and to describe the most frequent variants described in Italian patients.
Authors’ reply:
We are very grateful for the comments. As the reviewer mentioned, the differential diagnosis is very important for avoid contra-productive treatments, we do not insert a new paragraph, but we have modified the «Treatment» section (lines 148-150) to clarify and remark the importance of the diagnostic.
We follow the suggestions of the reviewer and we have added the recommended reference (Ferro E et al. FTL c.-168G>C Mutation in Hereditary Hyperferritinemia Cataract Syndrome: A New Italian Family. Pediatr Dev Pathol. 2018 Sep-Oct;21(5):456-460. doi: 10.1177/1093526618755200. Epub 2018 Feb 9. PMID: 29426274) in the «Introduction» section (lines 54-57)
Reviewer 2 Report
The work is a short review of the hereditary hyperferritinemia cataract syndrome, with the description of two families with known FTL mutations. The work is well written, although not original, since various reviews on the HHCS disorder have been written before, also by the same group. In addition, the two FTL mutations have been described before. More important is the rigorous mutant indication that follows the Human Genome Variation Nomenclature shown in fig 1 and in supplementary table.
- In fig 2 they may add the recent information that eIF4F binds IRE (Khan & Domashevskiy, 2021, doi: 10.1371/journal.pone.0250374)
- The main characteristic of HHCS is hyperferritinemia, thus it woud be important to show the serum ferritin levels of all the family members, particularly the affected ones, shown in figure 3
- The proposed first hypothesis of cataract formation based on ROS formation (line 114) should be supported by a reference, that is missing.
- The sentence «The HHCS patients do not have any other clinical symptoms rather than bilateral cataracts» (line 134) should be corrected in «The HHCS patients have no clinical symptoms other than bilateral cataract».
Author Response
IJMS/2021/05/17
REVIEWERS ANSWERS FIRST VERSION
Original Title and authors
Hereditary Hyperferritinemia Cataract Syndrome: Ferritin L Gene and Physiopathology Behind the Disease. Report of New Cases
Ferran Celma Nos 1,†, Gonzalo Hernández 1,†, Xènia Ferrer-Cortès 1,2, Ines Hernandez-Rodriguez 3, Begoña Navarro-Almenzar 4,5, José Luis Fuster 5, Mar Bermúdez Cortés 5, Santiago Pérez-Montero 2, Cristian Tornador 2 and Mayka Sanchez 1,2,*
† These authors contributed equally to this work.
Detailed responses to the reviewers’ comments and suggestions
We are grateful to the two reviewers for their thorough assessment of our paper, for their constructive input and for their overall very positive appreciation of our work. We have carefully revised our manuscript and addressed each point that was raised to our best, as detailed below. The new version of our manuscript contains the modifications as highlighted text for a rapid inspection.
Reviewer 1 Comments for the Author
The work is a short review of the hereditary hyperferritinemia cataract syndrome, with the description of two families with known FTL mutations. The work is well written, although not original, since various reviews on the HHCS disorder have been written before, also by the same group. In addition, the two FTL mutations have been described before. More important is the rigorous mutant indication that follows the Human Genome Variation Nomenclature shown in fig 1 and in supplementary table.
- In fig 2 they may add the recent information that eIF4F binds IRE (Khan & Domashevskiy, 2021, doi: 10.1371/journal.pone.0250374)
- The main characteristic of HHCS is hyperferritinemia, thus it would be important to show the serum ferritin levels of all the family members, particularly the affected ones, shown in figure 3
- The proposed first hypothesis of cataract formation based on ROS formation (line 114) should be supported by a reference, that is missing.
- The sentence «The HHCS patients do not have any other clinical symptoms rather than bilateral cataracts» (line 134) should be corrected in «The HHCS patients have no clinical symptoms other than bilateral cataract».
Authors’ reply:
We are very thanking to the reviewer for the feedback. We answer the suggestions in the respective order:
We kindly have modified Figure 2 according to the suggestions and we also added the corresponding reference (Khan & Domashevskiy, 2021, doi: 10.1371/journal.pone.0250374). lines (111-117)
We agree with reviewer that serum ferritin levels are relevant. We already had the serum ferritin values of the probands in Table 1. Additionally, we have included the serum ferritin levels in the main text for all of the three probands (lines 159, 189 and 208).
We agree with the reviewer, the reference was not in the right position. We are very sorry for the confusion and we correct the references (lines 122 and 124).
We agree with the reviewer and we have changed the original sentence by the following sentence in the manuscript (line 140): «The HHCS patients have no clinical symptoms other than bilateral cataract»